Host migration and environmental temperature influence avian haemosporidians prevalence: a molecular survey in a Brazilian Atlantic rainforest

http://orcid.org/0000-0001-7635-5279 Rodrigues Raquel A. 1
http://orcid.org/0000-0002-3901-4333 Felix Gabriel M.F. 2
http://orcid.org/0000-0002-9340-9010 Pichorim Mauro 3
http://orcid.org/0000-0002-6020-449X Moreira Patricia A. 4
http://orcid.org/0000-0001-5550-7157 Braga Erika M. 1 embraga@icb.ufmg.br
1 Parasitology, Universidade Federal de Minas Gerais , Belo Horizonte , Brazil
2 Animal Biology, Universidade de Campinas , Campinas , Brazil
3 Botany and Zoology, Universidade Federal do Rio Grande no Norte , Natal , Brazil
4 Biodiversity, Evolution and Environment, Universidade Federal de Ouro Preto , Ouro Preto , Brazil
Edwards Scott
Electronic publication date: 2021 Jun 22
Publication date: 2021
Volume: 9
Electronic Location ID: e11555
Received 2020 Sep 29; Accepted 2021 May 11
Copyright: © 2021 Rodrigues et al.
Copyright year: 2021
Copyright holder: Rodrigues et al.
License: This is an open access article distributed under the terms of the Creative Commons Attribution License, which permits unrestricted use, distribution, reproduction and adaptation in any medium and for any purpose provided that it is properly attributed. For attribution, the original author(s), title, publication source (PeerJ) and either DOI or URL of the article must be cited.
License URL: https://creativecommons.org/licenses/by/4.0/

Keywords: Plasmodium, Haemoproteus, Host-parasite interaction, Avian malaria, Parasite diversity

Funding: Fundação de Amparo à Pesquisa do Estado de Minas Gerais–FAPEMIG Conselho Nacional de Desenvolvimento Científico e Tecnológico–CNPq 474945/2010-3, 248588/2013-3 Coordenação de Aperfeiçoamento de Pessoal de Nível Superior (CAPES) This work was supported by Fundação de Amparo à Pesquisa do Estado de Minas Gerais–FAPEMIG; Conselho Nacional de Desenvolvimento Científico e Tecnológico–CNPq (M.P., grant numbers 474945/2010-3, 248588/2013-3), and Coordenação de Aperfeiçoamento de Pessoal de Nível Superior (CAPES). The funders had no role in study design, data collection and analysis, decision to publish, or preparation of the manuscript.

==============================
Avian haemosporidians are parasites with great capacity to spread to new environments and new hosts, being considered a good model to host-parasite interactions studies. Here, we examine avian haemosporidian parasites in a protected area covered by Restinga vegetation in northeastern Brazil, to test the hypothesis that haemosporidian prevalence is related to individual-level traits (age and breeding season), species-specific traits (diet, foraging strata, period of activity, species body weight, migratory status, and nest shape), and climate factors (temperature and rainfall). We screened DNA from 1,466 birds of 70 species captured monthly from April 2013 to March 2015. We detected an overall prevalence (Plasmodium/Haemoproteus infection) of 22% (44 host species) and parasite’s lineages were identified by mitochondrial cyt b gene. Our results showed that migration can be an important factor predicting the prevalence of Haemoproteus (Parahaemoproteus), but not Plasmodium, in hosts. Besides, the temperature, but not rainfall, seems to predict the prevalence of Plasmodium in this bird community. Neither individual-level traits analyzed nor the other species-specific traits tested were related to the probability of a bird becoming infected by haemosporidians. Our results point the importance of conducting local studies in particular environments to understand the degree of generality of factors impacting parasite prevalence in bird communities. Despite our attempts to find patterns of infection in this bird community, we should be aware that an avian haemosporidian community organization is highly complex and this complexity can be attributed to an intricate net of factors, some of which were not observed in this study and should be evaluated in future studies. We evidence the importance of looking to host-parasite relationships in a more close scale, to assure that some effects may not be obfuscated by differences in host life-history.

Introduction

The avian haemosporidians of the genera Plasmodium and Haemoproteus are vector-borne parasites that infect a wide range of host species (Hellgren, Pérez-Triz & Bensch, 2009; Ricklefs et al., 2014) and have frequently switched to new host species and new environments throughout their evolutionary history (Ricklefs, Fallon & Bermingham, 2004; Valkiūnas, 2005; Ricklefs et al., 2014; Ellis et al., 2019). These intracellular parasites reproduce sexually in different dipteran vectors: mosquitoes (Culicidae) are vectors of Plasmodium, and biting midges (Ceratopogonidae) and hippoboscid flies (Hippoboscidae) are vectors of Haemoproteus (Parahaemoproteus) and Haemoproteus (Haemoproteus), respectively (Valkiūnas, 2005; Santiago-Alarcon, Palinauskas & Schaefer, 2012). Concerning their vertebrate hosts, Plasmodium and Haemoproteus (Parahaemoproteus) infect birds of various orders, while Haemoproteus (Haemoproteus) is more specific and infects birds of the order Columbiformes and some sea birds (Work & Rameyer, 1996; Valkiūnas, 2005; Padilla et al., 2006; Levin & Parker, 2012; Levin et al., 2012).

Haemosporidians are closely connected to their hosts in interaction with outcomes ranging from sublethal effects on the host fitness (Ortego et al., 2008; Knowles, Palinauskas & Sheldon, 2010) to the decline and extinction of populations (van Riper et al., 1986; Atkinson et al., 1995, 2000). These parasites can exert selective pressure on host populations through effects on reproductive success, lifetime, and survival (Hamilton & Zuk, 1982; Scott, 1988; Spencer et al., 2005; Asghar et al., 2015; Ricklefs et al., 2016). Therefore, identifying the geographical distribution, host preferences, and infection prevalence of these parasites may help the development of appropriate management strategies to promote biodiversity conservation efforts worldwide.

Haemosporidian parasite prevalence, distribution, and richness vary widely across host species and can be affected by several factors. Prevalence may increase with age of host, since older individuals tend to have higher infection risk as a result of accumulated exposure to parasites or potentially immunosenescence (Atkinson et al., 1995; Ricklefs et al., 2005; Wood et al., 2007; Eastwood et al., 2019). However, this relationship is not consistent among studies and is not always observed, which may be indicative that it depends on the parasite and host species studied (Wood et al., 2007; Antonini et al., 2019). Shape and height of birds nest may also influence haemosporidian prevalence, nest height must be associated with the spatial feeding preferences of vectors when seeking hosts, while its shape must determine the birds’ exposure to vectors (Cerný, Votýpka & Svobodová, 2011; Fecchio et al., 2011; González et al., 2014; Lutz et al., 2015; Matthews et al., 2015).

A migration strategy is another factor that may also influence haemosporidian prevalence in birds, since migratory species are exposed to different vectors and parasites as a consequence of their habitat change during their annual cycle (Waldenström et al., 2002; Hellgren et al., 2013; Ricklefs et al., 2017; Slowinski et al., 2018; Pulgarín-R et al., 2019; Soares, Latta & Ricklefs, 2019). This relationship does not occur evenly between parasites and host species, as can be seen in studies that have observed that migratory birds can be infected by parasites from their wintering grounds (Waldenström et al., 2002) or have little or no haemosporidian lineage shared with resident birds (Ricklefs et al., 2017; Pulgarín-R et al., 2019). Likewise, the prevalence may be higher in migratory bird breeding (Hellgren et al., 2013) or wintering areas (Pulgarín-R et al., 2019).

Environmental factors can also be related to prevalence in a bird community. Seasonality, which involves temperature and rainfall, may influence vector infection dynamics and hence the haemosporidian transmission (Medeiros et al., 2016; Ferreira Junior et al., 2017; Hernández-Lara, González-García & Santiago-Alarcon, 2017a), and was even considered an important driver of host specialization (Fecchio et al., 2019). However, seasonality, in the same way as the host life-history traits, is not always linked to haemosporidian prevalence, not even in the tropics (Ishtiaq, Bowden & Jhala, 2017). Moreover, as seasonality is also associated with the breeding period of birds, a relationship could be observed between these two variables influencing the haemosporidian prevalence, since during bird’s breeding season may be an increase in adult susceptibility to infections, due to changes in its behavior and immunity (Drobney, Train & Fredrickson, 1983; Richner, Christe & Oppliger, 1995; Ardia, 2005). Besides that, juveniles born throughout this time may also be more susceptible to infections (Cosgrove et al., 2008; Møller, 2010; Santiago-Alarcon et al., 2011; Ferreira Junior et al., 2017; Rodrigues et al., 2020). Therefore, many factors interact to determine the parasite prevalence in birds, revealing distinct results in studies testing the same relationships in different communities. This makes the knowledge of these interactions in different ecosystems valuable, to better understand the various factors that interact and influence the dynamics of parasite-host infection in natural environments.

Here, we tested whether the probability of an individual being infected with haemosporidian parasites is related to (1) its individual-level traits (i.e., age and breeding condition), which may influence the host immune defense mechanisms and individual exposure to vectors; (2) its species-specific traits (i.e., diet, foraging strata, period of activity, species body weight, migratory status, and nest shape), which may be linked to differential vector exposure; and (3) climate factors (i.e., temperature and rainfall), which might influence in vector abundance and richness. These hypotheses were tested in an area of Brazilian Restinga, a poorly sampled phytophiosionomy for hemoparasites, and therefore with a high potential to house new haemosporidian lineages.

Materials & methods

Study site

The study was carried out in Barreira do Inferno Rocket Launch Center of the Brazilian Air Force (acronym in Portuguese, Centro de Lançamento Barreira do Inferno–CLBI; 5°55′S 35°9′W), a protected area of ~1,800 ha located in Parnamirim, State of Rio Grande do Norte, northeastern Brazil (Fig. 1). The area is in a tropical coastal vegetation region, named Restinga, which is a type of vegetation associated with the Atlantic Rainforest Domain (for more details about the study site, see Rodrigues et al. (2020)). Following the local climatic data (Fig. S1), we determined as rainy season the period that goes from March to August and, as dry season the period from September to February.

Figure 1 Map of the study site sampled for a bird community infected by Plasmodium and Haemoproteus, Barreira do Inferno Rocket Launch Center of the Brazilian Air Force, Parnamirim, State of Rio Grande do Norte, Brazil.

We sampled 36 to 49 points per month in a cyclical way so that all points were sampled one time at every three month. These three sample areas are indicated by the different shades of gray in the image—two areas with 36 sample points and one area with 49 sample points.

Bird sampling

We monitored monthly the haemosporidian infection in a bird community within a 30 ha plot (550 m × 550 m) from April 2013 to March 2015. The birds were captured following the protocol described in Rodrigues et al. (2020). All captured birds were identified and banded with individual aluminum bands provided by CEMAVE/ICMBio (permission 3239). Recaptured birds were not used in this study. We evaluated birds for age (adult or young) based on their plumage, labial commissure, and cranial ossification; and the presence of brood patch, by visual examination of the birds. We obtained blood samples through brachial venipuncture with a sterile needle (13 × 4.5 mm) and stored the blood in filter paper kept at 4 °C until DNA extraction.

Our use of mist-nets and banding was approved by the Brazilian biodiversity monitoring agency (Institute Chico Mendes for Biodiversity Conservation—ICMBio, Brazilian National Center for Bird Conservation—CEMAVE, permit 3239). We followed standard ethical protocols for wildlife animals. Time handling the birds was kept to the minimum, and all birds were released after banding, data, and sample collection. This study was approved by the Ethics Committee in Animal Experimentation (CETEA), Universidade Federal de Minas Gerais, Brazil (Protocol #254/2011).

Parasite detection

The parasite detection followed the protocols described in Rodrigues et al. (2020). Lineages without previous records in the database were considered new lineages and deposited in GenBank under accession numbers MH260577, MK291501, MK291502, MK291503, MK291504, MK291505, MK291506, and MK291507. New occurrences of sequences previously described were also deposited in GenBank under accession numbers MK264392, MK264393, MK264394, MK264395, MK264396, MK264397, MK264398, MK264399, MK264400, MK264401, MK264402, MK264403, MK264404, MK264405, MK264406, MK264407, MK264408, MK264409, MK264410, and MK264411.

Modeling the factors predicting avian malaria prevalence

To test which factors influence the haemosporidian prevalence in the bird community, we modeled the individual probability of infection separately for each parasite genus (Plasmodium and Haemoproteus), as a function of individual and species-level traits of hosts, as well as the climatic conditions. We chose to model the individual probability of infection rather than the prevalence in host populations (see Fecchio et al., 2013) since we have predictor variables at the individual level, which included age of host (adult or young) and breeding condition (breeding or non-breeding). The breeding condition was determined by the presence of a brood patch at the time of the capture. Species-specific traits included average body mass (g), migratory status (migratory or resident), nest shape (open-cup vs closed), diet (frugivore, nectarivore, granivore, insectivorous, omnivore, or combinations of two or three diets), foraging stratum (ground, understory, midheight, canopy, and combinations of two or three foraging strata), and period of activity (day or night). We obtained the species-specific traits data from Handbook of the Birds of the World Alive (Hoyo, Elliott & Christie, 2011) and, from Wilman et al. (2014). The classification of the diet was made considering the food item or the combination of food items that covered at least 80% of the total diet, based on the data from Wilman et al. (2014), using at most three main food items per species of bird. We considered as migratory the species of birds with some type of seasonal displacement in the area, following the classification used by Somenzari et al. (2018). Finally, we used mean monthly temperature and total monthly precipitation for the climatic variables (through the 24 months study), which were centered and scaled before analysis. We obtained the climate data from a Brazilian Meteorological Database for Education and Research (INMET, 2017). Because in some cases we were unable to obtain all the individual-level traits in the field, we removed from the dataset individuals with missing information to run the analysis.

We accounted for different types of pseudoreplication (Hurlbert, 1984) in our dataset. Phylogenetically related bird species, for instance, would have similar infection probabilities (Ricklefs & Fallon, 2002; Waldenström et al., 2002), as well as individuals captured at the same occasion (same month and same seasonality; Kim & Tsuda, 2010; Ferreira Junior et al., 2017). Therefore, to control for such potential dependences, we used the R package lme4 (Bates et al., 2015; R Core Team, 2016; Harrison et al., 2018) to fit Generalized Linear Mixed Models (GLMM; Bolker et al., 2009). The infection status was recorded as a binary response variable (0: uninfected; 1: infected) and modeled as a Bernoulli trial with a binomial distribution of errors and the logit link function. The hosts individual- and species-level traits, and the climatic conditions entered as fixed factors, without any interactions; and the temporal (season and month) and phylogenetic (order, family, gender, species) factors entered as random factors–nested within each dimension (temporal: month nested in season; phylogenetic: species nested in the genus, and genus nested in the family) and crossed among dimensions. Here we used data referring to the taxonomic classification of birds, considering that this is similar to the currently known phylogenetic classification. The random effects were modeled affecting only the intercept, but not the slope of the model.

The final model was obtained by backward selection of the fixed factors only–the random structure was maintained complete in all models (Barr et al., 2013). Starting from the full model, we used the likelihood ratio test to remove the fixed factors that do not contribute significantly to the model fit (Crawley, 2013). The likelihood ratio test compares the data likelihood under the full model against the data likelihood under a model with fewer factors and was performed using an analysis of variance (ANOVA) performed by the anova function. In each step, we removed the fixed factors that explained the small part of the deviance. We used the r.squaredGLMM function implemented in the R package MuMIn (Barton, 2018) to compute both the marginal and conditional R2 for the final model; and the icc function implemented in the R package sjstats to compute the adjusted intraclass-correlation of the random factors. The marginal-R2 gives the percentage of variance explained by the fixed factors, while the conditional-R2 gives the total percentage of variance explained by the full model, including the fixed and the random factors (Nakagawa & Schielzeth, 2013). Finally, the adjusted intraclass correlation gives the percentage of the residual variance explained by each random factor (Nakagawa, Johnson & Schielzeth, 2017). The overdispersion test was not necessary, because an overdispersion test does not make sense with a binary response variable (Crawley, 2013). All these analyses were made separately for each parasite genus (i.e., Plasmodium and Haemoproteus), but we included in the dataset only host species with at least one individual infected by the parasite genus that was being analyzed on each occasion.

Results

Overall malaria prevalence

We captured 1,466 individual birds of 25 families and 70 bird species, of which 322 (22%, 44 species) were infected by Plasmodium/Haemoproteus. All samples that screened positive were subjected to the cytochrome b PCR, which successfully amplified infections from 145 individuals. We obtained high-quality sequences from 117 samples. This is a well-established methodology for detecting haemosporidians that has been successfully applied in many other studies (e.g. Lacorte et al., 2013; Fecchio et al., 2017a; Ferreira Junior et al., 2017; Ricklefs et al., 2017; Ferreira et al., 2020; Lopes et al., 2020; Rodrigues et al., 2020; Soares, Young & Ricklefs, 2020), allowing us to observe the parasite prevalence and richness in a host community and compare the identified lineages with other haemosporidian studies around the world. Unfortunately, we were not able to collect and analyze blood smears from the captured birds, which would greatly enrich our findings and allow us to assess the discrepancy between the number of positive samples and the number of successfully sequenced lineages. We detected Plasmodium in 35 individuals of 18 species (Table 1) and Haemoproteus infections were detected in 67 individuals of 15 species (Table 2). Among more highly-captured species (n ≥ 7), the highest prevalence of infection were detected in Cyclarhis gujanensis (n = 9/14, 64.3%), Tachyphonus rufus (n = 69/108, 64%), and Columbina passerina (n = 11/21, 52.4%). The majority of bird species caught is resident in the region, but we have captured 15 migratory species of which nine were infected (Elaenia spp. [3 species; n = 69/431], Myiarchus tyrannulus [1/4], Turdus amaurochalinus [23/164], Schistochlamys ruficapillus [4/13], Turdus flavipes [3/13], Cyanerpes cyaneus [1/1] and Vireo chivi [5/10]; Table S1).

Table 1 Distribution of Plasmodium lineages across bird species captured in Barreira do Inferno Rocket Launch Center of the Brazilian Air Force, Parnamirim, State of Rio Grande do Norte, Brazil.

Bird species	Plasmodium lineages		
	BAFLA03	BAFLA04	CALON01	CPCT57	DENPET03	FOGRI01	FOMEL04	H012	HYAMA01		
Cantorchilus longirostris (25)			2								
Coereba flaveola (114)		3			3						
Cyanocorax cyanopogon (2)								1			
Cyclarhis gujanensis (14)											
Elaenia chilensis (244)	1	1									
Elaenia spectabilis (9)		1									
Formicivora grisea (10)						1					
Formicivora melanogaster (9)							1				
Herpsilochmus pectoralis (24)											
Herpsilochmus sellowi (27)											
Hylophilus amaurocephalus (29)									1		
Leptotila verreauxi (9)		1									
Piaya cayana (8)				1							
Polioptila plúmbea (15)											
Tachyphonus rufus (108)		1									
Turdus amaurochalinus (156)					1						
Turdus flavipes (11)					1						
Turdus leucomelas (98)											
	Plasmodium lineages		
Bird species	LECOR02	PADOM09	PADOM11	PADOM17	PAMIT01	POPLU01	TUAMA01	TURNUD02	U12	Total	
Cantorchilus longirostris (25)										2	
Coereba flaveola (114)		1								7	
Cyanocorax cyanopogon (2)					1					1	
Cyclarhis gujanensis (14)					1					1	
Elaenia chilensis (244)		1		1				1		5	
Elaenia spectabilis (9)										1	
Formicivora grisea (10)										1	
Formicivora melanogaster (9)										1	
Herpsilochmus pectoralis (24)	1									1	
Herpsilochmus sellowi (27)	1									1	
Hylophilus amaurocephalus (29)										1	
Leptotila verreauxi (9)										1	
Piaya cayana (8)										1	
Polioptila plúmbea (15)		1				1				2	
Tachyphonus rufus (108)			1							2	
Turdus amaurochalinus (156)							1		1	3	
Turdus flavipes (11)		1								2	
Turdus leucomelas (98)					1					1	
Note:

The number of individuals captured for each species is denoted in parentheses.

Table 2 Distribution of Haemoproteus lineages across bird species captured in Barreira do Inferno Rocket Launch Center of the Brazilian Air Force, Parnamirim, State of Rio Grande do Norte, Brazil.

		Haemoproteus lineages	
Bird species	Família	COTAL01	ELALB01	NYMAC01	PAPOL03	SocH3	SocH4	TARUF02	UN203	VIREO02	Total	
Coereba flaveola (114)	Thraupidae							1			1	
Columbina passerina (21)	Columbidae					4	1				5	
Columbina talpacoti (14)	Columbidae	4									4	
Coryphospingus pileatus (5)	Thraupidae							1			1	
Cyclarhis gujanensis (14)	Vireonidae							1	5		6	
Elaenia chilensis (244)	Tyrannidae		1						1		2	
Formicivora grisea (10)	Thamnophilidae							1			1	
Myiarchus tyrannulus (4)	Tyrannidae		1								1	
Neopelma pallescens (27)	Pipridae							1			1	
Nystalus maculatus (10)	Bucconidae			3							3	
Pachyramphus polychopterus (7)	Tityridae				2						2	
Tachyphonus rufus (108)	Thraupidae							37			37	
Tangara cayana (67)	Thraupidae							1			1	
Turdus flavipes (11)	Turdidae							1			1	
Vireo chivi (10)	Vireonidae									1	1	
Note:

The number of individuals captured for each species is denoted in parentheses. SocH3, SocH4 and COTAL01 are Haemoproteus (Haemoproteus) lineages. ELALB01, NYMAC01, PAPOL03, TARUF02, UN203 and VIREO02 are Haemoproteus (Parahaemoproteus) lineages.

Haemosporidian diversity

We recovered 27 cyt b lineages from 117 individuals, of which 18 were Plasmodium lineages, detected in 35 birds (18 species), and 9 were Haemoproteus lineages, detected in 67 birds. Fifteen of the 117 high-quality sequences exhibited multiple infections, based on double peaks in the chromatograms, and were removed from the dataset. Among the Haemoproteus lineages, 3 were Haemoproteus (Haemoproteus) lineages detected in 9 birds (2 species), and 6 were Haemoproteus (Parahaemoproteus) lineages detected in 58 birds (13 species), as shown in Table 2. A total of eight lineages were detected here for the first time (five Plasmodium and three H. (Parahaemoproteus)). Of the 27 lineages, 14 (52%) were detected only once. Of the 13 lineages detected at least twice, eight (30%) were found in more than one host species (Table 1 and Table 2). Most of the lineages detected in only one host species were found in only one individual (14 lineages), and the remaining lineages were found in two (2 lineages), three (1 lineage), or four (2 lineages) individuals.

Despite the higher richness of Plasmodium lineages detected in birds, there was a higher number of birds infected by Haemoproteus (35 and 67 birds, respectively), mainly by H. (Parahaemoproteus). The most prevalent Plasmodium lineage was BAFLA04, detected in 7 birds, being four captured at the rainy season and three at the dry season. Among the Parahaemoproteus lineages, we highlight the lineage TARUF02, which was detected here for the first time in 44 birds (21 at the rainy season and 23 at the dry season) and have an apparent preference in infecting birds of the species Tachyphonus rufus. Of the total 44 birds infected by this lineage, 37 were T. rufus species, and the other seven occurrences of this lineage were detected in seven different bird species (Table 1 and Table 2). Besides, we recorded only two infections by other lineages in T. rufus species (PADOM11 and BAFLA04).

Factors predicting haemosporidian prevalence

Our final dataset for prevalence analysis, after excluding individuals with missing data, included 1,187 individual birds. Although the subgenus Haemoproteus (Haemoproteus) and H. (Parahaemoproteus) are classified within the same genus, they are very different concerning their vertebrate hosts and vectors (Valkiūnas, 2005), which made us consider it important to treat these two groups differently in our study. However, as the number of birds infected by H. (Haemoproteus) was too low (n = 9), we only included in these analyses birds infected by H. (Parahaemoproteus).

The GLMM analysis indicated that the temperature influences the probability of infection by Plasmodium (Table S2), with the increase of one standard deviation on temperature (0.99 °C) resulting in an increase of 1.8 in the odds of infection by Plasmodium (the odds ratio, Table 3). However, the temperature is not an important factor influencing the probability of infection by Parahaemoproteus and neither Plasmodium nor Parahaemoproteus were influenced by rainfall.

Table 3 The parameters of the minimal models (binomial GLMMs) explaining the probability of infection by Plasmodium sp. and Parahaemoproteus sp.

Parasite genus	Main effects	Estimate	Std. Error	Odds ratio	Z value	P value	R2m	R2c	
Plasmodium	Intercept	−2.85	0.32	0.05	−8.69	2e−16*	0.08	0.24	
	Temperature	0.59	0.21	1.80	2.73	0.0060*			
Parahaemoproteus	Intercept	−0.70	0.19	0.49	−3.60	0.0003*	0.43	0.43	
	Migratory	−3.37	0.54	0.03	−6.42	4.32e−10*			
Note:

Both models have the same random structure (see Tables S1–S3).

The probability of infection was not influenced by any of the individual host traits tested, such as age and breeding condition for both parasite genus. Besides, when considering species-specific traits, only the probability of infection by Parahaemoproteus was affected by the species migratory status (Table S3), with the odds of be infected by Parahaemoproteus in migratory birds being 0.03 of that in non-migratory birds. However, there was no influence on the probability of infection by any haemosporidian when considering diet, foraging stratum, period of activity, species body mass, and nest shape.

For Parahaemoproteus GLMM the ICCadj was equal to zero for all random factors (Table S4), and, consequently, marginal and conditional R2 were identical (R2m = R2c = 0.43, Table 3). This means that the random factors do not explain anything about the dispersion of model residuals. In this case, the GLMM collapse to a simple GLM. For Plasmodium GLMM, otherwise, the ICCadj was also equal to zero for almost all random factors, except for random factor ‘species’ which was equal to 0.17 (Table S4), indicating that 17% of the residual variance is correlated within species. Thus, marginal and conditional R2 were not identical (R2m = 0.08, and R2c = 0.24, Table 3) and, therefore, the species of the birds had a small influence on the probability of Plasmodium infection. This indicates that, apart from the other factors, birds of different species will have different probabilities of becoming infected by Plasmodium.

Discussion

What does influence avian haemosporidian prevalence? Here, we found that migratory birds were less likely to be infected with Haemoproteus (Parahaemoproteus) when compared to resident birds and that the probability of infection by Plasmodium was positively influenced by temperature. By observing avian haemosporidians in a diverse region, and exploring how ecological variables are related to parasite infection probability in wild birds, we can compare these interactions we have found here with patterns of interactions already observed in avian communities in different contexts, to add knowledge that allows us to better understand infectious diseases in wild birds.

Migratory behavior of birds from CLBI had a significant and inverse association with the probability of infection by Haemoproteus (Parahaemoproteus), but there was no effect of migration on the probability of infection by Plasmodium. The current knowledge of the host specificity of the parasite lineages predicts that Haemoproteus parasites tend to be more host-specific than Plasmodium parasites (Ishtiaq et al., 2007, 2010; Dimitrov, Zehtindjiev & Bensch, 2010). Therefore, it is possible that migratory birds are not suitable hosts for Parahaemoproteus lineages with which they geographically overlap during their annual cycle. This agrees with the study presented by Hellgren et al. (2007) showing that Haemoproteus and Leucocytozoon had a significant affiliation to a single resident bird fauna, while Plasmodium lineages showed a higher degree of infecting both resident and migratory bird species. If that is true in the studied bird community, migrants may be less likely to become infected by Haemoproteus in CLBI when compared to resident birds, due to the greater specificity this parasite presents concerning its local hosts. Because Plasmodium lineages are usually more generalist, they can probably infect migrants as well as resident birds, which would explain why we did not find the same influence of migratory behavior on the probability of infection by these parasites. If, on the one hand, the migration has the potential to increase the exposure of birds to parasites by concentrating individuals at breeding, overwintering or, migratory stopover sites (Waldenström et al., 2002), on the other hand, migration could make possible the escape of birds from habitats where parasite transmission stages have accumulated, or selective removal of infected hosts during movements (Hall, Altizer & Bartel, 2014).

Alternatively, our results might be related to the hypothesis that the selection experienced by migratory birds in their breeding and wintering areas resulted in greater investment in immune defense (Møller & Erritzøe, 1998). If a bird individual is negative for haemosporidian infections, it might be because either the host individual is not infected, the parasite is dormant in tissues and not found in the bloodstream (Valkiūnas, 2005), or that it occurs in such low intensities in the avian blood that it is not detected by PCR screening. This former situation could be a cue that the bird was able to fight the infection, reducing its parasitemia. In that case, migrants having greater investment in immune defense could also have low parasitemia when infected.

Despite the absence of data about mosquitoes in CLBI to allow us to correctly evaluate the vector-parasite-host relationship, the positive association observed between temperature and Plasmodium prevalence may be related to the effects of temperature on the vectors of this parasite. For avian haemosporidians, the temperature is commonly described as an important abiotic factor influencing the parasite development and vector breeding opportunities (Beier, 1998; Santiago-Alarcon, Palinauskas & Schaefer, 2012; Medeiros et al., 2016; Mordecai et al., 2019). It has been demonstrated that the development of different malaria parasites in vectors can be influenced by the climate and is generally hampered by low increments in temperature (LaPointe, Goff & Atkinson, 2010; Zamora-Vilchis, Williams & Johnson, 2012). Temperature also determines the rate at which mosquitoes develop into adults, the frequency of their blood-feeding, and the rate at which parasites are acquired (Patz et al., 2000). Garamszegi (2011) has shown that a 1 °C increase in global temperature led to a two- to three-fold increase in the average prevalence of Plasmodium in birds. It is also demonstrated that studies made with samples from years and localities where temperature anomalies were strongly expressed generally detected higher Plasmodium prevalence than surveys based on samples that were less affected by temperature anomalies (Garamszegi, 2011). Sehgal et al. (2011), in a study conducted on Olive Sunbirds (Cyanomitra olivacea) in West and Central Africa, also showed an association between higher temperatures and elevated Plasmodium prevalence, with data indicating that the maximum temperature of the warmest month was the most important indicator for elevated malaria prevalence. In contrast, Zamora-Vilchis, Williams & Johnson (2012), in a study conducted in Australia, demonstrated that in areas with high temperatures the birds had a higher prevalence of Haemoproteus, and relationships for Leucocytozoon and Plasmodium were also positive but not statistically significant. Given that, in a warmer climate, the vector abundance may increase, and the transmission of vector-borne diseases must be higher. On the other hand, in our study the infection probability by Parahaemoproteus lineages was not influenced by temperature and, it is possible that biting midges, which act as their vectors, do not have their reproduction and development as closely related to climatic factors as Plasmodium vectors. Immature biting midges require a certain amount of free water or moisture, being able to develop in a wide range of habitats that meet that criterion like pools, streams, marshes, bogs, beaches, swamps, tree holes, irrigation pipe leaks, saturated soil, animal dung, and even rotting fruit and other vegetation (Mellor, Boorman & Baylis, 2000). With so many possible breeding sites, Parahaemoproteus vectors are probably present in the community during the whole year and, even if there is some reduction in their abundance due to changes in temperature and rainfall (Mellor, Boorman & Baylis, 2000), it may be less evident for biting midges than for mosquitoes.

Neither age nor breeding condition explained the probability of infection by haemosporidians. Although many studies have evidenced that birds age and/or breeding condition may influence haemosporidian prevalence (Wood et al., 2007; Ferreira Junior et al., 2017; Hernández-Lara, González-García & Santiago-Alarcon, 2017b; Eastwood et al., 2019), others have failed to detect such an association (Ricklefs et al., 2005; Matthews et al., 2015). There was also no association between prevalence and rainfall or, except for migratory behavior, any tested species-specific traits (diet, foraging strata, period of activity, species body weight, and nest shape). It is true that many studies have shown that different host-traits and abiotic factors are important determinants in a host-parasite interaction (Ricklefs et al., 2005; Wood et al., 2007; Medeiros et al., 2016; Ferreira Junior et al., 2017; Hernández-Lara, González-García & Santiago-Alarcon, 2017a; Ishtiaq, Bowden & Jhala, 2017; Fecchio et al., 2017b; Eastwood et al., 2019). However, there are many variations in these studies’ results, and some of them fail to detect these interactions. Based on the mixed results found in these studies, it is possible that the relationship between species-specific traits as well as individual-level traits and the risk of infection by haemosporidian parasites might be location-dependent. It is important to highlight that several factors might be working together to determine such variations we see in all these different studies, including the host species that, as we observed, had a small influence on the infection probability by Plasmodium. That small influence of the species on Plasmodium prevalence could be related to factors that were not tested in our studies, like phylogeny or co-infection with other hemoparasites. Studying haemosporidian infections only at the community level reduces our ability to detect if some species are more susceptible to the most common Plasmodium lineages in CLBI, for example. That is why species-specific studies are also important and allow us to identify some relationships that may be overshadowed in a bird community (Rodrigues et al., 2020).

The CLBI harbors a diverse community of avian haematozoan lineages distributed among 63% (44/70) of the bird species sampled in this study and we estimated an overall parasite prevalence of ~22%. Estimates for the prevalence of haemosporidian parasites in bird communities from Brazil indicate a great variation both among different ecosystems and between different sites in the same ecosystems. The estimated prevalence in Cerrado varied from 21% to 42% (Belo et al., 2011; Fecchio et al., 2013; Lacorte et al., 2013). In other Brazilian habitats it has also been observed a great variation in prevalence estimates, e.g., 17.4% (Fecchio et al., 2017c) to 21.7% (Svensson-Coelho et al., 2013) in Amazonian Region; 38.5% (Lacorte et al., 2013) to 42% (Ferreira Junior et al., 2017) in Seasonally Dry Tropical Forest; and 12.4% to 39.6% in Atlantic Forest (Ribeiro et al., 2005; Sebaio et al., 2010; Lacorte et al., 2013). This considerable variation in prevalence among studies is evidence that we still have many aspects of this complex, spatially variable parasite–host system to understand. The avian haemosporidian parasite–host community in northeast Brazil adds to our understanding of the distribution and diversity of avian haemosporidian parasites and examines the ecological factors that influence host susceptibility. However, we acknowledge that much remains to be investigated in the parasite-host relationship in Restinga and suggest that future studies use information from blood smears and mixed infections to extend the ability to detect and identify haemosporidians in this bird community.

Conclusions

In this first exploration of avian haemosporidian parasites in a largely unexplored region of Brazil, we could demonstrate that this environment harbors a high diverse community of Plasmodium and Haemoproteus parasites. We established that migration can be an important factor predicting the prevalence of Haemoproteus, but not Plasmodium, in hosts. Thus, in CLBI, Haemoproteus lineages infect preferably resident birds and should be more difficult to disperse into new environments. The other individual- and species-level traits were not important in determining the probability of infection by Plasmodium or Haemoproteus in our study, which indicates a great variation of the influence of these factors on haemosporidian prevalence in different communities. The temperature, but not the rainfall, seems to predict the Plasmodium prevalence in this bird community. This result raises the possibility that ongoing climate change will impact the dynamics of Plasmodium transmission, a subject that should be explored in future studies. The higher number of birds infected by Haemoproteus lineages than by Plasmodium is an uncommon finding in Brazil and led us to suggest that northeast Brazil must have a different haemosporidian infection dynamics when compared to other studied regions of the country. Further investigations in northeast biomes and sampling of the haemosporidian vectors are needed to better understand the transmission dynamics and to elucidate the factors promoting higher levels of Haemoproteus infection in birds of this region.

Supplemental Information

Supplemental Information 1 Migratory birds captured in a bird community in Barreira do Inferno RocketLaunch Center of the Brazilian Air Force, Parnamirim, State of Rio Grande do Norte,Brazil.

The infection status, and parasites’ genus and lineages, when identified, are provided. The sample size and number of infections detected in each bird species, and the number of birds infected by a parasite lineage are denoted in parentheses. Because not all diagnosed infections were sequenced, the number of birds in the “Infection Status” column may not be equivalent to the number in the “Lineage” column.

Click here for additional data file.

Supplemental Information 2 Backward selection of the main effects influencing the probability of infection by Plasmodium sp.

The full random structure was maintained in all models. The Plas.Glmm1 model starts with all main effects (fixed factors), which were sequentially removed. In each step we removed the main effect that explained the small amount of deviance (indicated on the fixedfactor column). The procedure was repeated until no effect could be removed without causing significant loss of model fit (indicated by the p value column). In this case, this happened only when temperature was removed from the model.

Click here for additional data file.

Supplemental Information 3 Backward selection of the main effects influencing the probability of infection by Haemoproteus (Parahaemoproteus) sp.

The full random structure was maintained in all models.The Plas.Glmm1 model starts with all main effects (fixed factors), which were sequentially removed. In each step we removed the main effect that explained the small amount of deviance (indicated on the Fixed Factors column). The procedure was repeated until no effect could be removed without causing significant loss of model fit (indicated by the p value column). In this case, this happened only when migratory status was removed from the model, although theimprovement in deviance provided by Body mass and Breeding period is marginally significant, indicating a possible role of these factors in the probability of infection by Haemoproteus (Parahaemoproteus) sp.

Click here for additional data file.

Supplemental Information 4 Adjusted intra-class correlation (ICCadj) of temporal and taxonomic factors.

The factors were considered as two dimensions of the random structure of the models built to analyze the factors predicting the probability of infection by Plasmodium and Haemoproteus in a bird community in Barreira do Inferno Rocket Launch Center of the Brazilian Air Force, Parnamirim, State of Rio Grande do Norte, Brazil.

Click here for additional data file.

Supplemental Information 5 Climatogram representing the average monthly temperatures and total monthly precipitation during the period of study in Barreira do Inferno, Parnamirim, State of Rio Grande do Norte, Brazil.

Click here for additional data file.

Supplemental Information 6 Raw data.

Click here for additional data file.

We thank the Centro de Lançamento Barreira do Inferno-CLBI for giving us access inside the protected area to sample birds. We thank the Brazilian National Center for Bird Conservation for providing the aluminum bands to mark captured birds. We also thank all LabOrnito—UFRN ornithologists and students for their valuable help in the field and laboratory work, especially Lidiane M. Andrade and Priscilla S. A. Araújo. We are grateful to Eric Pereira for designing the map of our sampling area.

Additional Information and Declarations

Competing Interests

Author Contributions

Animal Ethics

Field Study Permissions

DNA Deposition

Data Availability

Érika M. Braga is an Academic Editor for PeerJ.

Raquel A. Rodrigues conceived and designed the experiments, performed the experiments, analyzed the data, prepared figures and/or tables, authored or reviewed drafts of the paper, and approved the final draft.

Gabriel M.F. Felix analyzed the data, prepared figures and/or tables, authored or reviewed drafts of the paper, and approved the final draft.

Mauro Pichorim performed the experiments, authored or reviewed drafts of the paper, and approved the final draft.

Patricia A. Moreira conceived and designed the experiments, performed the experiments, authored or reviewed drafts of the paper, and approved the final draft.

Erika M. Braga conceived and designed the experiments, performed the experiments, authored or reviewed drafts of the paper, and approved the final draft.

The following information was supplied relating to ethical approvals (i.e., approving body and any reference numbers):

This study was approved by the Ethics Committee in Animal Experimentation (CETEA), Universidade Federal de Minas Gerais, Brazil (Protocol #254/2011).

The following information was supplied relating to field study approvals (i.e., approving body and any reference numbers):

Our use of mist-nets and banding was approved by the Brazilian Biodiversity Monitoring Agency (Institute Chico Mendes for Biodiversity Conservation—ICMBio, Brazilian National Center for Bird Conservation—CEMAVE, permission 3239).

The following information was supplied regarding the deposition of DNA sequences:

Lineages without previous records in the database were considered new lineages and are available in GenBank under accession numbers MH260577, MK291501, MK291502, MK291503, MK291504, MK291505, MK291506 and MK291507.

New occurrences of sequences previously described are also available in GenBank: MK264392, MK264393, MK264394, MK264395, MK264396, MK264397, MK264398, MK264399, MK264400, MK264401, MK264402, MK264403, MK264404, MK264405, MK264406, MK264407, MK264408, MK264409, MK264410 and MK264411.

The following information was supplied regarding data availability:

The raw data for host age, breeding condition, and presence of ectoparasites of birds are available as Supplemental Files.

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
