# Peer review of "Host migration and environmental temperature influence avian haemosporidians prevalence: a molecular survey in a Brazilian Atlantic rainforest"

_PeerJ, doi:10.7717/peerj.11555_

## Round 0.1 · original submission · Major Revisions

The two reviews are very thorough and both give very useful suggestions for improving the manuscript. Reviewer 1 has some specific comments and reviewer 2 feels that the paper should be toned down in its claims, increase self-criticism a bit, and attend to some data analysis details. Overall, however, the reviewers are excited about this paper.

Reviewer 1 ·

Basic reporting

Brazil has tremendous biodiversity, and many understudied ecosystems. With over 1800 bird species, it is one of the nations richest in avifauna. Within these birds are their blood parasites, of which, with molecular methods, the scientific community is discovering a large number of undescribed species. At this point, in the MalAvi database, hundreds of haemosporidian cytochrome b lineages have been described from Brazil. Many of these were identified by the group of Erika Braga, who presides over the work presented here in the manuscript “Host migration and temperature influence avian haemosporidians prevalence: a molecular survey in a Brazilian Atlantic Rainforest”. The work describes the parasites of over 1400 birds tested from the Restinga habitats of Northeast Brazil. These coastal forests are threatened by human development, and they harbor some endemic bird species. Of the 70 species of birds studied here, the authors found the overall prevalence of Haemoproteus and Plasmodium to be 22.3%. Interestingly, they found that compared to resident birds, migratory birds have a lower prevalence of Haemoproteus. They also found an association of Plasmodium prevalence with temperature. However, other traits, such as age, presence of ectoparasites and breeding condition were not associated with parasite prevalence. They identified 8 new cyt b parasite lineages. The large-scale sampling regime is particularly impressive. The work is relatively straightforward and will be recognized as providing baseline prevalence data for birds of this region of Brazil. I have a few comments that would strengthen this manuscript.

Experimental design

1. It would be important to mention why blood smears were not made in this study. Providing data from blood smears would help with the description of new species, and also contribute information about parasitemia. Perhaps due to limited resources, it was not possible to make blood smears, but this should be discussed.

2. The beginning of the Results section is somewhat confusing. The authors found 322 individuals infected with Plasmodium or Haemoproteus but in the next sentence “All samples that screened positive were subjected to the cytochrome b PCR, which successfully amplified infections from 145 individuals”. It would be important to clarify why the two different PCR methods should yield such different results. Perhaps in the Discussion, you could point out why you chose to use two different PCR assays, and the advantages of each. Can you actually trust that the first 322 were actually positive? Please provide information about how many times the initial PCR was done, just once, or repeatedly. This type of information is directly pertinent to the validity of prevalence data. Also, in line 274, 15 of the 322 infected host individuals (4.6%) exhibited multiple infections. In fact it would be more correct to state that 15 of the 102 sequenced individuals were multiple infections, since the prevalence varies so widely with the PCR methods. Another reason to collect blood smears is to verify PCR results on prevalence.

3. In line 303, it is not clear why the number of birds decreased from 1443 to 1187 for the final dataset. Was it because these were recaptures, or was there not enough blood? Please clarify.

4. Temperature can vary widely at the microclimate level. I suspect that the INMET temperature data would be very broad. In the future, in taking temperature measurements, it would be more helpful to take more localized temperature readings. In any case, it is difficult to evaluate how temperature affects the prevalence, because in most cases, the haemosporidian infections are chronic, meaning that the bird will harbor the parasite for many months, or years. It is difficult to determine in what season the birds were infected. For these reasons, it would be helpful in the Discussion to go into more depth relating the data here to previous studies where temperature correlated to Plasmodium prevalence. You already include the references by Ishtiaq, 2017 and LaPointe, 2010. There are some more that are pertinent to malaria transmission in general. For example:
a. Sehgal, R. N. M., et al. "Spatially explicit predictions of blood parasites in a widely distributed African rainforest bird." Proceedings of the Royal Society B: Biological Sciences 278.1708 (2011): 1025-1033.
b. Mordecai, Erin A., et al. "Thermal biology of mosquito‐borne disease." Ecology letters 22.10 (2019): 1690-1708.
c. Fecchio, Alan, et al. "Climate variation influences host specificity in avian malaria parasites." Ecology letters 22.3 (2019): 547-557.

5. Similarly, it would be helpful to have more lengthy discussion about how the results with migration, age, breeding condition and ectoparasites corroborate or differ from previous studies.

6. Lines 283-287: These relationships between Plasmodium and Haemoproteus have been established many times. This can be omitted.

7. In general, the English writing could be improved. I will leave this to the editors, but a couple spelling mistakes: Line 391- whole (not hole). Line 81-spatial (not spacial).

8. Figure 1: Please explain in the Figure Legend the different colors of green shading in the “Sampling Area”.

9. Figure 2: It would be very helpful to include well-characterized morphospecies in this phylogenetic tree. For example, P. relictum, P. circumflexum…. and also several of the well described Haemoproteus lineages. This would help put this tree into context with other known studies about phylogenetic inferences of these parasites.

10. Typically these types of studies also include Leucocytozoon lineage data. You could briefly discuss why those data were not included in this work.

Validity of the findings

no comment

Additional comments

In sum, this study adds to the literature about the overall lineage diversity of haemosporidian parasites. The sampling was thorough, and the number of birds tested is impressive. It would be helpful to put the work more into context by describing the special characteristics of the Restinga habitats, and the rarity of the samples. Although the use of the different PCR methods is somewhat confusing, and a lack of blood smears problematic, the work provides important information about the diversity of these parasites in a threatened region of Brazil.

Reviewer 2 ·

Basic reporting

English is in general good, but several parts need rewriting.

A few extra references could be added in 2-3 specific places mentioned below.

Professional article structure.

Self-contained and relevant results.

Experimental design

Everything appropriate except that some parts of the methods need to be improved.

Validity of the findings

The finding that migration is an important factor is based on an odds-ratio of 0.03, which leaves space for criticism.

I required an improvement of the final sentence of the Discussion and a broader approach than only Brazil.

Additional comments

The manuscript provides information about avian haemosporidians from a novel region the restinga of northeast Brazil. Total sample size is good collected from dozens of species during a two-year period. The manuscript is in general well written, has an almost adequate cover of the literature and provides new insights about the theme. Several parts of the Methods need clarification. Though I liked the study, several aspects need consideration.

First, my major criticism is that the manuscript needs self-criticism about its limitations. It would benefit from a self-criticism about the small sample sizes for several species (large sample sizes were highly biased to a few species), or some potential limitations about the methods. For example, by using brood patch as an index of breeding activity all species in which only one sex (usually the female) develop a brood patch, on average 50% of the individuals of that species would be wrongly assigned as not breeding. Also, all migratory individuals will necessarily not be in breeding condition at the wintering sites. Explain how these shortcomings might have affected you results. Furthermore, ectoparasitism was classified as either yes or no, but most birds in any region will have a very low load of ectoparasites, thus with potential low effect on hosts. A better approach would be to use ectoparasitism intensity either as a continuous variable or as in classes (ex., zero, small load, heavy load). Thus, ectoparasite presence/absence might not have been a good method, and this should also be addressed in the Discussion.

Second, the final argument in the Discussion (lines 422-426) is too strong considering the limitations above. Also, it is contrary to the several studies cited on lines 399-402, and in the third paragraph of the Introduction. At the most, this argument could be used for this study site.

Further comments:

1) The title has double meaning as written since temperature' can be interpreted either as host temperature or environmental temperature. Also, restinga, though a vegetation typical of the Atlantic Forest biome, is not a true rainforest. It would be more appropriate to use something like "Brazilian Atlantic xeric forest".
2) Line 82. It is more common to use nest type (open-cup vs. closed) than nest shape.
3) Line 89. “…, for example, some …” needs rewriting.
4) Line 100. “Moreover, seasons can also influence the prevalence in birds concerning reproduction”, is awkward and needs rewriting.
5) Lines 100-102. “Moreover, seasons can also influence the prevalence in birds concerning to reproduction, since during bird’s breeding season may be an increase in adult’s susceptibility to infections, due to changes in its behavior and immunity”. This assertion needs a reference.
6) Lines 109-116. In this paragraph, insert the 'area of Restinga' in the beginning of the paragraph, or add a new sentence at the end informing the reader that the whole study was conducted at a Restinga site.
7) Lines 136-142. Invert the order of this paragraph with the next.
8) Line 140. For birds it is much more common to use “banding” than “tagging”. Same in line 151.
9) Line 152. Provide an explanation of how you evaluated birds for age (adult or young).
10) Line 153. Explain which stage of the brood patch (full patch?) was considered as breeding or not breeding condition.
11) Line 204. It seems that you refer to species and not population specific traits.
12) Line 206. Use only open-cup x closed.
13) Lines 210-212. Explain how you classified the died of each species. Did you follow HBW or Wilman et al. (2014)? How did you access the number of food items? It is unclear as written.
14) Lines 214-216. This part is unclear. Did you use mean temperature for which period? The average of the two years of the study? The month of capture? Same for precipitation?
15) Lines 251-252. It is unclear why species not infected were excluded. They could be used to test hypotheses 2 and 3.
16) Line 262. I would not consider a sample of > 7 as 'well', but simply as better.
17) Lines 266-268. The sample sizes of some of these species do not match the numbers in the tables (Table 1, Table 2, Table S1).
18) Line 275. You removed individuals with double infections. Justify the reason for the exclusion of these individuals.
19) Lines 318 and paragraph start at line 344. Consider what an odds-ratio of 0.03 really means. What is its biological meaning?
20) Lines 349-350. “Therefore, it is possible that migratory birds are not suitable hosts for Parahaemoproteus lineages with which they geographically overlap during their annual cycle.” Do you have evidence from other similar studies that could provide support for this argument? Provide a reference.
21) Lines 387-388. “… that meet that criterion breeding sites, ...” This sentence is incomplete.
22) Line 405. “…including the very host-species that …” This is unclear.
23) Lines 416 and 443. Why mention only Brazil? The manuscript should have an appeal for a broader scientific community, at least neotropical. It would be much more interesting and useful to mention the neotropical region.
24) Line 431. It is not clear in the text that this is a high diverse habitat, of if you refer to high diversity of birds or of parasites. If you mention birds, provide some reference.
25) Table 2. Check the names of Elaenia albiceps and E. chilensis in the tables. There seems to be some error.

---

## Round 0.2 · Minor Revisions

The reviewer finds the paper improved, but still suggests more refinement of the English language in the text. Can you find someone to read over the text with you to improve it? This would be the best strategy, in addition to addressing the other minor points of the reviewer.

Reviewer 1 ·

Basic reporting

I was pleased to read the revised manuscript entitled “Host migration and environmental temperature influence avian haemosporidians prevalence: a molecular survey in a Brazilian Atlantic Rainforest”. I believe that the authors addressed all of the reviewers’ comments regarding the original manuscript in a succinct and transparent way. I agree that the phylogenetic tree was largely unnecessary, and the authors now better explain the potential limitations of the study. The figures are more clearly presented, and all the numbers now make sense.

I would only suggest that the authors again go through the manuscript with a critical eye for the English writing. I have found several instances that I would recommend for correction (below I present the corrected version of the sentences). However, I think that there are still sentences that I may have missed that could be improved.

Title- avian haemosporidian prevalence

23- and are considered a good model to study host-parasite interactions.

30- and parasite lineages

41- at a closer scale

46- vector-borne

49- Valkiūnas (throughout the manuscript)

61- to the decline

95- Moreover, as seasonality

95- associated with

102- parasite prevalence

210- An overdispersion test

228- findings and allow us to assess the discrepancy between the number of positive samples…

231- Among more highly-captured species…

382- these studies’ results

390- infections only at the community level..

Experimental design

no comment

Validity of the findings

no comment

Additional comments

This is a nice study from a very unusual region of Brazil, and I believe the data will be useful to future scientists studying birds and parasites in relation to climate change.

---

## Round 0.3 · accepted · Accept

Thank you for incorporating the reviewer's comments.